# A National Survey of Dispensing Practice and Customer Knowledge on Antibiotic Use in Vietnam and the Implications

**DOI:** 10.3390/antibiotics11081091

**Published:** 2022-08-12

**Authors:** Thuy Thi Phuong Nguyen, Thang Xuan Do, Hoang Anh Nguyen, Cuc Thi Thu Nguyen, Johanna Catharina Meyer, Brian Godman, Phumzile Skosana, Binh Thanh Nguyen

**Affiliations:** 1Department of Pharmaceutical Management and Economics, Hanoi University of Pharmacy, Hanoi City 10000, Vietnam; 2The National Center of Drug Information & Adverse Drug Reaction Monitoring, Hanoi University of Pharmacy, Hanoi City, 10000, Vietnam; 3Bach Mai Hospital, Hanoi City 10000, Vietnam; 4Department of Public Health Pharmacy and Management, School of Pharmacy, Sefako Makgatho Health Sciences University, Pretoria 0204, South Africa; 5Department of Pharmacoepidemiology, Strathclyde Institute of Pharmacy and Biomedical Sciences, University of Strathclyde, Glasgow G4 0RE, UK; 6Centre of Medical and Bio allied Health Sciences Research, Ajman University, Ajman P.O. Box 346, United Arab Emirates

**Keywords:** antibiotics, dispensing, self-purchasing, drug retailers, knowledge, regulations, Vietnam

## Abstract

Misconceptions and pressures have increased the sales of antibiotics without a prescription across countries. There are concerns with such practices in Vietnam given rising antimicrobial resistance rates. A national survey was conducted among 360 private drugstores located in nine provinces in Vietnam. Anonymous interviews were conducted with participants selected by convenience sampling. Subsequently, multivariable logistic regression analyses were undertaken evaluating the relationship between customer characteristics and antibiotic purchases. A total of 480 out of 1626 surveyed participants purchased antibiotics, 81.7% of which did not have a prescription, involving 29 different antibiotics. In 86.4% of these, participants were prescribed antibiotics by drug sellers. Most antibiotics were sold to treat respiratory tract infections (61.4%), with the ‘Access’ antibiotics (amoxicillin and cephalexin) being the most frequently sold. Only one-fifth of participants understood that they were breaking the law by purchasing antibiotics without a prescription. Participants purchasing antibiotics without a prescription had lower awareness concerning antibiotic laws and treatment duration (*p* < 0.05). Under 50% agreed to having a doctors’ prescription in the future when purchasing antibiotics. Freelancer occupation (OR = 0.52, 95% CI = 0.83–0.96) and a lower educational level (OR = 0.49, 95% CI = 0.25–0.96) were factors related to purchasing antibiotics without a prescription. Overall, we recommend increasing fines and monitoring of drugs stores, greater promotion of the family doctor system as well as increasing media and educational campaigns to limit self-purchasing of antibiotics in Vietnam and reduce resistance.

## 1. Introduction

The advent of antibiotics has ushered in a new era in the treatment of infectious diseases, significantly decreasing morbidity and mortality associated with common infectious diseases [1]. However, these benefits are being affected by their inappropriate use, which increases antimicrobial resistance (AMR) [2]. AMR is a worldwide problem already resulting in a high number of deaths, which were estimated to reach 4.95 million deaths globally in 2019 [3], and will potentially result in up to 10 million deaths annually by 2050 unless addressed [4,5]. Alongside this, AMR use may potentially result in considerable costs of up to USD 3.4 trillion annually by 2030 if not addressed, equivalent to 3.8% of gross global domestic product [6,7]. The purchasing of antibiotics without a prescription, particularly for self-limiting conditions such as upper respiratory tract infections (URTIs), is a key factor contributing to the misuse of antibiotics and increasing AMR in most low- and middle-income countries (LMICs) [8,9,10]. Whilst there are laws prohibiting the purchasing of antibiotics without a prescription and warnings in the media, dispensing antibiotics without physician prescriptions is still common practice in many LMICs, with variable implementation of these laws [9,10,11,12]. This is a concern that needs to be urgently addressed, especially in LMICs.

Healthcare administration in Vietnam is presently organized into a three-level system, mirroring the division of healthcare facilities [13]. Currently, a prescription is compulsory by law in Vietnam for an antibiotic to be dispensed, and an antibiotic should only be dispensed by a qualified pharmacist, who must always be available in pharmacies during opening times. However, despite these strict regulations and constant supervision from the pharmaceutical regulatory authority, the sale of antibiotics without a prescription is still widespread among community pharmacists in Vietnam, similar to other LMICs [9,10,11,14,15,16,17,18,19]. This is fuelled by currently limited sanctions for community pharmacists in Vietnam when dispensing antibiotics without a prescription, with a fine of only USD 15–25 per documented violation when caught [20]. Initially, a fine of 100 USD–150 USD was introduced in 2005 (Decree 45/2005/NĐ-CP) [20,21], which was subsequently decreased to 15 USD–25 USD in 2013 (Decree 176/2013/NĐ-CP) [20]. 

Alongside this, in Vietnam, co-payments are typically only applied in hospitals by the Vietnam Health Insurance once patients have had their medical treatment and/or examination [22]. This may result in extensive co-payments once the cost of medicines is included, which may end up costing the patient considerably more than just purchasing medicines directly from community pharmacies. Furthermore, going straight to pharmacists is typically more convenient and does not require wasting time waiting to see a physician, which may also appreciably reduce the daily wage [11].

According to the Pharmaceutical Law of Vietnam, pharmacies should be managed by pharmacists with a bachelor’s degree and only operating in urban zones [23,24,25]. However, pharmacies only accounted for approximately 30% of 61,867 drug retailers among 64 provinces nationwide in 2018 in Vietnam [26]. The remaining proportion are drug stores and drug counters, which are managed by pharmacists with a middle college diploma and principally concentrated in rural areas [23,24]. However, sales of antibiotics without a prescription have contributed up to 88% of total sales in urban areas and 91% in rural areas with the self-medication proportion comprising up to 50% in urban regions [18,19]. The differences in qualifications, as well as the extent of monitoring of community pharmacies and drug stores, raises issues and challenges in addressing current concerns with the potential of increasing AMR and associated morbidity, mortality and costs [11,27].

In order to reduce inappropriate practices among community pharmacies and drug stores, and any potential risks for patients, effective interventions are needed. These include increased education of both patients and pharmacists as well as monitoring of community pharmacies, sanctions and realistic fines [11,12,28,29]. This is especially true in the case of Vietnam, where there has been an appreciable increase in the number of pharmacists in recent years alongside high alert regarding AMR across both community and hospital settings, which is causing concern among the medical authorities. Potentially effective interventions for any given circumstances can come from surveys evaluating the current situation. This can be combined with appraising the situation in other LMICs, including activities and their impact, to reduce the self-purchasing of antibiotics and provide guidance. Evaluating the current status was the rationale behind undertaking point prevalence studies regarding antimicrobial use in hospitals across countries [30]. The findings may subsequently be used to plan future quality improvement programs. However, previous studies conducted among community pharmacies and drug stores in Vietnam aimed at assessing the extent of purchasing of antibiotics without a prescription have been hindered by typically only focusing on a single local region and with limited sample sizes [16,18,19,31]. In addition, these studies typically focused only on the sales of antibiotics without a prescription rather than total antibiotics dispensed. Consequently, we aimed to address this gap by comprehensively investigating antibiotic dispensing practices on a national scale in Vietnam as well as evaluating customer knowledge on antibiotic use. Customers’ knowledge is seen as crucial since they are a key driver of antibiotic prescribing and dispensing in ambulatory care, especially for self-limiting conditions, such as URTIs [9,15,32]. The findings can be used to direct the authorities in Vietnam regarding potential appropriate and effective interventions to reduce the extent of inappropriate dispensing of antibiotics in the community, building on the experiences of other countries, especially LMICs (Table 1).

In view of this, the objectives of this study were to undertake a comprehensive survey of current antibiotic dispensing practices across Vietnam including both community pharmacies and drug stores. In addition, we intended to ascertain customers’ knowledge regarding antibiotics including the current legal situation concerning the dispensing of antibiotics without a prescription and factors affecting AMR. As mentioned, the combined findings can be used to suggest potential ways forward for the authorities in Vietnam given the reduced value of potential fines in 2022 following their substantial decrease in 2013.

## 2. Results

### 2.1. Study Population

Out of 1626 participants observed and interviewed, most of them (86.8%) purchased medicines without a prescription, with 28.2% purchasing antibiotics. Out of 487 participants purchasing antibiotics, 81.7% (398 out of 487) purchased antibiotics without a prescription, with the remaining antibiotics being dispensed with a prescription. Among participants purchasing antibiotics, 215 purchased antibiotics for themselves while 136 purchased them for others, primarily for children aged 6 months to 12 years or relatives. Most antibiotics (86.4%) were recommended by drug sellers based on customers’ reported health conditions, with customers requesting an antibiotic explicitly making up the remainder (13.6%) (Figure 1).

### 2.2. Demographic Characteristics of Participants Purchasing Antibiotics 

Regarding the demographic characteristics of participants purchasing antibiotics, the majority were female (64.6%), freelance workers (62.2%) and had an educational level below college or university (67.1%) (Table 2). Comparing those who purchased with or without a prescription, those participants purchasing antibiotics without a prescription were typically males, had lower education levels and were freelance workers. The difference between groups was statistically significant in terms of education and occupation (*p* < 0.05).

### 2.3. Pathological Classification for Purchased Antibiotics

Both groups, i.e., those with and those without a prescription, mainly purchased antibiotics for problems relating to respiratory tract infections (61.4%), followed by the digestive system (13.1%) and eye-related symptoms (6.0%) (Table 3). Among respiratory-related infections, most purchasing antibiotics without a prescription did so to treat flu-like symptoms, coughs, fever, colds, sore throats, rhinitis, sinusitis or bronchitis. The most common symptoms were influenza (17.4%), followed by a sore throat (16.6%) and a cough (16.6%).

### 2.4. Type of Antibiotics Dispensed

The types of antibiotics dispensed for those without a prescription were more varied than those with a prescription, i.e., 29 different antibiotics compared with 25 different antibiotics, respectively. Based on the WHO AWaRE list, 10 out of 25 antibiotics in the group with a prescription were in the ‘Access’ group, with the remainder in the ‘Watch’ group and none in the ‘Reserve’ group. Among the 29 different antibiotics purchased without a prescription, 12 were from the ‘Access’ group, 17 from the ‘Watch’ group and none from the ‘Reserve’ group. Over 90% of antibiotics supplied in both groups were oral formulations. The most common antibiotics purchased by those with a prescription were cefuroxime (20.4%), followed by co-amoxiclav (12.6%) and levofloxacin (7.8%). In contrast, the five most common antibiotics purchased by the group without a prescription were cephalexin (25.3%), amoxicillin (25.6%), cefuroxime (9.2%), ciprofloxacin (6.2%) and co-amoxiclav (5.9%) (Figure 2).

### 2.5. Knowledge of Customers about Antibiotics

Only 21.1% of the participants knew that purchasing antibiotics without a prescription is currently illegal in Vietnam (Table 4), with a statistically significantly better awareness among those with a prescription versus those without a prescription (*p* = 0.032). As a result, 48.6% of participants agreed in the future to have a doctors’ prescription when purchasing antibiotics, with a statistically significantly higher proportion of those being individuals who had a prescription (63.8%) versus those purchasing an antibiotic without a prescription (44.2%; *p* < 0.000). Among the participants disagreeing to purchase antibiotics with a doctors’ prescription, those with a college or university educational level had the highest percentage of disagreement (23.2%, twice as high as the remaining groups), with the difference between groups being statistically significant. By contrast, there was no statistically significant difference between different occupational groups (*p* = 0.136).

Participants purchasing antibiotics without a prescription refused to visit doctors to obtain a prescription beforehand because they thought that their disease or symptoms were mild (50.5%) and that they would save time (30.6%) and money (12.3%) by not visiting doctors for their condition. Additionally, personnel in the drug stores had often treated their diseases or symptoms beforehand (15.4%), and they had experience with successfully treating their current condition (14.2%).

The proportion of respondents with correct answers about the treatment duration of antibiotics was 40%; i.e., antibiotics need to be taken for at least five days or as long as recommend by a physician, even if the recipient begins feeling better. Of concern is that nearly 20% of participants reported that antibiotics should only be used for a short time (under two days). Participants with a prescription showed a significantly better awareness than those without a prescription about treatment duration (50.6% versus 37.6%, respectively; *p* = 0.026).

Regarding AMR, over 50% of participants in both groups had heard of, or knew about, antibiotic resistance. Despite this, only 48.7% knew that not taking antibiotics for long enough leads to AMR. However, more participants (52.8%) agreed that AMR was a severe public health problem, with no statistical difference between those with or without a prescription when purchasing antibiotics.

### 2.6. The Relationship between Participants’ Demographic Characteristics and Their Purchasing of Antibiotics

The univariate and multivariate logistic regression models in Table 5 indicate that two influential factors regarding the purchasing of antibiotics without a prescription were the occupation of the participants and their educational level. Overall, non-freelance workers (OR = 0.52, 95% CI = 0.83–0.96) and those who had attended universities (OR = 0.49, 95% CI = 0.25–0.96) had a lower tendency of purchasing antibiotics without a prescription.

## 3. Discussion

To the best of our knowledge, we believe this is the first comprehensive study to investigate drug retailer habits and customers’ knowledge regarding antibiotics and current regulations across all nine provinces in Vietnam. With the large sample sizes and the selection of drug retailers across a number of provinces in Vietnam, the study was able to provide a comprehensive view of current practices and participants’ knowledge about antibiotics to guide future activities and policies.

The results demonstrate that the purchase of antibiotics without a prescription is still common practice across Vietnam despite current regulations (81.7% of participants stated they would still purchase antibiotics without a prescription). These findings were similar or higher than recent results in China (45.5–88.4%) [17,45], Cameroon (47%) [46] and earlier in Saudi Arabia (48.4–70.7%) before greater enforcement of the laws and fines [28,47] as well as Sri Lanka (30.2%) [29]. However, our results show appreciably greater dispensing of antibiotics without a prescription in Vietnam than seen in Saudi Arabia following enforcement of their regulations coupled with appreciable fines for abuse [28], and in the Republic of Srpska [43] (Table 1). Similarly, versus those seen in Shaanxi Province, China following their multiple initiatives [38] (Table 1). However, the rates seen in our study were lower than those seen in some prior studies in Vietnam (83.3–91%) [18,31], although higher than others (55.2%) [16]. In addition, the rates seen in our study were lower than seen in Zambia, where at one stage 100% of pharmacies surveyed dispensed antibiotics without a prescription [11]. The differences seen across countries might be due to differences in socio-cultural-economic circumstances, the healthcare administration system, including monitoring of community pharmacists, as well as the extent and follow-up of potential fines and other sanctions (Table 1). The accessibility of healthcare professionals, including physicians and pharmacists, as well as the level of co-payments for both physicians and medicines, may also have been a factor. Within a country, differences in the findings might arise from the sample size of the investigations (limited to one province) [18,31], the method in which the antibiotic quantities were estimated (by the number of purchasers or antibiotics sold), the data collection method (direct customer interviews or direct observations) and the type of pharmacies surveyed [44]. In addition, the timing of the research and whether there had been recent tightening of the regulations as seen, for instance, in Brazil, Mexico, Saudi Arabia and the Republic of Srpska [28,43] (Table 1), may also have been factors. Alongside this are the qualifications of the pharmacy personnel interviewed, since the presence of trained pharmacists has reduced self-purchasing of antibiotics, especially for viral infections, in some countries (Table 1) [29,41,48].

Antibiotics were mainly purchased to treat RTIs (61.4%), similar to findings across countries including China (45.5%), Saudi Arabia in 2018 prior to recent initiatives and Sri Lanka [11,29,45,47]. As seen, the main respiratory-related symptoms for which antibiotics were dispensed were flu-like symptoms, coughs and colds. This is a concern as antibiotics are generally unnecessary for these viral infections [11,49]. Whilst this is against current Vietnamese law, which prohibits the dispensing of antibiotics without a prescription, the high rate seen in our study could be explained by the current fines, which are limited to only USD 15–25 per violation if caught [20,21]. This compares with possible earnings that could be accrued by potentially dispensing multiple prescriptions without being caught. This is different to the situation in the Republic of Srpska and Saudi Arabia, where there were appreciably greater fines for violations along with greater enforcement of the regulations, which resulted in appreciable reductions in the extent of self-purchasing of antibiotics in both countries in recent years (Table 1). In addition, drug sellers might have gained appreciable experience with treating many patients, or imitated other drug sellers’ behaviour, so they are confident with their recommendation of an antibiotic in given circumstances even if this is illegal, especially if this satisfies the patient [50,51]. On the other hand, self-diagnosing and self-medication practices raise concerns about health-related complications, namely, irrational drug selection, uncontrolled drug reactions, AMR and drug overuse [11,31,44,52,53].

Two major influential factors regarding the purchasing of antibiotics in our study were low educational levels and freelance workers, similar to study findings in Cameroon and Saudi Arabia [46,47]. The purchasing of antibiotics in Vietnam without a prescription may also be driven by a high percentage of drug sellers fearing they would lose customers if they do not dispense an antibiotic until the regulations are fully enforced [18]. This is a concern with a greater number of ‘Watch’ antibiotics being sold without a prescription versus those with a prescription, accounting for nearly 20% of total antibiotic use in our study. This is facilitated by the high perceived effectiveness of broad-spectrum antibiotics and lower prices leading to frequent sales. However, there are concerns with the actual level of benefit of broad- versus narrow-spectrum antibiotics to treat common infections alongside more adverse effects [54]. In addition, using these antibiotics in the early stages of their lifecycles is a key driver of AMR [55]. As a result, educational programs aiming to increase drug sellers’ awareness of the key issues surrounding antibiotics and reduce their dispersion without a prescription are critical going forward [11,29,56,57].

A key issue was the limited knowledge that participants had regarding antibiotics, with only one-fifth of those surveyed knowing that antibiotics purchased without a prescription is illegal. This misplaced knowledge was possibly caused by drug sellers continuing to provide antibiotics without a prescription, resulting in participants believing that non-prescription sales of antibiotics were still permitted. Furthermore, a recent global study revealed that 57.6% of those surveyed did not know that medicines prescribed by pharmacists contained antibiotics [10]. There can also be disquiet about the knowledge of antibiotics among community pharmacists in LMICs; however, this is not always the case [11,29,57,58]. Alongside this, there are also concerns that insufficient instructions are being given to patients dispensed antibiotics with a high percentage of patients misunderstanding the duration of treatment (60%). This is a concern as reduced treatment lengths can increase AMR. Encouragingly though, most patients surveyed knew about AMR and its negative effect in the community. However, under 50% knew that not taking antibiotics for the agreed duration increased AMR.

As mentioned, the reasons for participants going to pharmacies and asking for antibiotics to be dispensed without a prescription included wasting time and money to see a physician and mild symptoms not warranting a visit to a physician as well-treated by drug sellers in the past. Similar reasons are seen across countries, especially where there are high co-payments for physician visits and where time and expense are factors in the decision to see a physician [11,15,44]. In addition, the effort needed by drug sellers to effectively dissuade patients that they do not need to purchase antibiotics for essentially viral infections taking up time is a further factor [15,51]. These factors and issues need to be addressed in Vietnam going forward to reduce current high AMR rates.

### Study Limitations

We are aware there are limitations with this study. Direct observation might have resulted in the percentage of non-prescribed consumption of antibiotics being lower than the reality because of the change or the adjustment in drug sellers’ behaviour. However, observations were conducted at many drug retailers across the country to reduce this potential bias. In addition, the large sample size used in our study, coupled with the inclusion of drug retailers across a number of provinces in Vietnam, should provide a comprehensive view of current practices and the customers’ knowledge about antibiotics. Compared to other prior studies, trained researchers in this study observed customers’ behaviour during the actual purchasing of antibiotics instead of directly asking them to reduce bias. Moreover, participants were interviewed directly afterwards instead of via a self-administrated questionnaire, and the interview process was undertaken directly outside the drug retailers to reduce bias. Consequently, missing information or recall errors should be limited. In view of this, we believe our findings are robust and provide guidance to the authorities in Vietnam regarding potential future activities to reduce high rates of self-purchasing antibiotics without a prescription.

## 4. Materials and Methods

### 4.1. Study Setting

The cross-sectional national study was conducted among 360 private drug retailers in nine provinces in Vietnam. Out of these, four had the largest number of central cities (Hanoi, Da Nang, Ho Chi Minh and Can Tho), and five were provinces representing different geographical locations (Thanh Hoa—North Central Coast, Khanh Hoa—South Central Coast, Dak Lak—Central Highlands, Binh Duong—Southeast and Kien Giang—Red River Delta).

A multistage sampling method was applied to recruit drug dispensers, including both community pharmacies and drug stores. Firstly, the local health authorities provided a list of all registered private drug retailers in each region. Subsequently, 20 pharmacies and 20 drugstores from each province were randomly selected using the random sampling function in Microsoft Excel. This included all licensed private drug retailers who were in operation and had a certificate of Good Pharmacy Practices as a requested standard criterion in the drug law of Vietnam. Other stores selling only herbal medicines and traditional medicines were excluded. Finally, 360 drug retailers were identified among the nine provinces (200 community pharmacies and 160 drug stores) for this study.

### 4.2. Ethical Considerations, Data Collection and Sample Population and Data Collection

Ethical approval for the study was granted by the Research Board of Hanoi University of Pharmacy (No 1116/QĐ-DHN dated 30 November 2016). In addition, the study was approved by the local health authorities, and consent to take part was also obtained from each drug retailer and participant in advance.

Data were collected over a period of two years by fifth-year pharmacy students and a number of the co-authors from the Hanoi University of Pharmacy, who were trained before the study began to enhance the output as well as make sure the data collectors were mindful of the sensitivities of the situation.

Inclusion criteria were any customer purchasing medicines from drug retailers during the study period that were not subsequently excluded. Exclusion criteria included anyone unwilling to participate, those under 18 years of age, non-Vietnamese as the questionnaire was only available in Vietnamese, those with a disability or those just purchasing herbal supplements to treat their condition rather than medicines (prescribed or self-purchased).

During this period, 1992 potential participants were identified from the 360 drug retailers. The median number of potential participants per pharmacy per day was 50 (IQR = 30–80) and 30 (IQR = 20–50)) for drug stores. Among the 1992 possible participants, 192 people refused to participate, and 174 were just purchasing dietary supplements; consequently, they were excluded from the study as they did not meet the inclusion criteria. A final sample of 1626 participants were subsequently included in the study.

In order to minimize changes in participants’ behaviours, data collectors first observed the potential participant when they first entered the drug retailer. They were subsequently invited to be interviewed as they left the store following their purchase. The observation process would be kept for data analysis if the customer agreed to participate; otherwise, these participants would be excluded from any analysis. Before potentially being interviewed, participants were informed about the background and purpose of the study. Confidentiality, voluntary participation and the right to withdraw from the study at any time during the interview were communicated to potential participants before starting the interview. If customers were subsequently willing to participate in the study, and over 18 years old, they were requested to provide written informed consent according to the Helsinki Declaration.

All face-to-face interviews with participants were undertaken outside of the drug retailers to help retain confidentiality and avoid distractions.

### 4.3. Study Questionnaire

A questionnaire was developed for the purpose of the study based on previous research [59,60,61] and adjusted by the authors for suitability for the Vietnamese context (Appendix A). After development, the questionnaire was pre-tested among ten customers and pharmacists to evaluate their understanding and acceptance of the questions. Following the pre-testing stage of the questionnaire, some of the questions were modified, which enhanced the validity and robustness of the questionnaire used in the principal study.

The questionnaire was structured into three parts in order to observe potential participants’ purchasing behaviour and subsequently assess their knowledge about antibiotics through the interview process as well as collect demographic information. The first section of the survey sought to understand current medicine utilization patterns as well as any information on each purchased antibiotic through observation by trained data collectors.

The second part contained multiple-choice and open-ended questions relating to participants’ antibiotic knowledge, i.e., the legal situation with respect to antibiotic dispensing, treatment duration of antibiotics, AMR and the level of acceptance in buying antibiotics without a prescription. This section consisted of four questions regarding participants’ knowledge of antibiotics answered as yes, no, unknown/not sure and two questions regarding their attitude to antibiotic use answered on a Likert 3-point scale (agree, disagree and partly agree) or unknown. Finally, the third part collected participants’ demographic characteristics. Responses were anonymized.

### 4.4. Data Management and Statistical Analysis

Antibiotics were classified by the Anatomical Therapeutic Chemical (ATC) Classification System [62], and subsequently by the WHO AWaRe antibiotic classification [63]. Regarding the WHO AWaRe antibiotic classification, the ‘Access’ group should be affordable, quality-assured and readily accessible. The ‘Watch’ group should be used for a limited number of indications due to high resistance potential, and the ‘Reserve’ group of antibiotics should only be used for infections of multi-resistant bacteria or those who have failed with other antibiotics [63,64].

Data were entered into Epidata 3.1 (Microsoft™), cleaned and encoded. R version 4.0.1. was used to perform descriptive and inferential statistical analysis of the data. The χ2 or Fisher’s exact tests were used to investigate differences in demographic characteristics and awareness between antibiotic groups. Multivariate logistic regression was used to identify the association between the dependent variable (non-prescribed or prescribed antibiotics) and independent variables (age, gender, occupation and education). A *p*-value of less than 0.05 was considered as statistically significant.

## 5. Conclusions and Recommendations

Over 80% of participants in our study purchased antibiotics without prescriptions mainly to treat ARIs, with only a small proportion of participants knowing that it was illegal to purchase antibiotics without a prescription. There were also concerns that participants were not fully aware of the optimum treatment duration for antibiotics and AMR-related issues. These findings suggest an urgent need for patient- and pharmacy-targeted interventions to address inappropriate antibiotic dispensing.

As mentioned, patient activities could include continued education regarding the optimal use of antibiotics and the limited need for antibiotics for essentially viral infections such as ARIs. This could be via social media and other channels given their growing influence across countries [65], and specifically aimed at freelance workers or those with low educational levels.

Activities aimed at drug retailers could include increasing their education to include greater input regarding antibiotics. Alongside this, the need for fully qualified pharmacists to be more knowledgeable and conversant with the appropriate management of patients with viral and other self-limiting infections including URTIs through improved university education and following graduation [11,29,48]. This should also include the ready availability of simple-to-use and trusted guidelines for the management of common infections in all community drug retail stores, which has been used to good effect in the Republic of Srpska [43].

Other activities aimed at drug retailers could include increased fines for citizens able to purchase antibiotics without a prescription for essentially viral infections. This could potentially be monitored via mobile technologies or other electronic means linked to the dispensing of antibiotics in retail stores, along with strengthening the monitoring of drug retailers in Vietnam. The reduction in the value of fines, especially from 2005 levels, has limited the impact of this measure in practice in Vietnam. This needs to be reversed, building on the impact of the multiple measures in the Republic of Serbia and Saudi Arabia including appreciably greater fines (Table 1).

Other suggestions include a greater promotion of the family doctor system in Vietnam, including the possibility of remote medical examinations. This builds on initiatives introduced during the recent COVID-19 pandemic, with subsequent electronic prescribing connected to the pharmacy system. This should be welcomed in more rural areas in Vietnam where physicians are less concentrated, and remote consultations should appreciably reduce co-payments and travel costs. Electronic prescribing also offers the potential for instigating quality targets centring on limiting the prescribing of antibiotics for essentially viral infections as well as promoting ‘Access’ versus ‘Watch’ antibiotics when needed. Alongside this, expanding health insurance payments to pharmacies to counter-act possible losses of income from dispensing antibiotics without a prescription, especially from the ‘Watch’ list, could be beneficial. We will be monitoring these suggestions in the future.

## Figures and Tables

**Figure 1 antibiotics-11-01091-f001:**
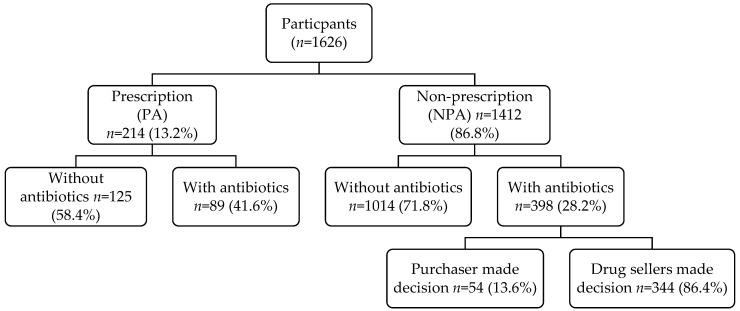
Flow chart of participants.

**Figure 2 antibiotics-11-01091-f002:**
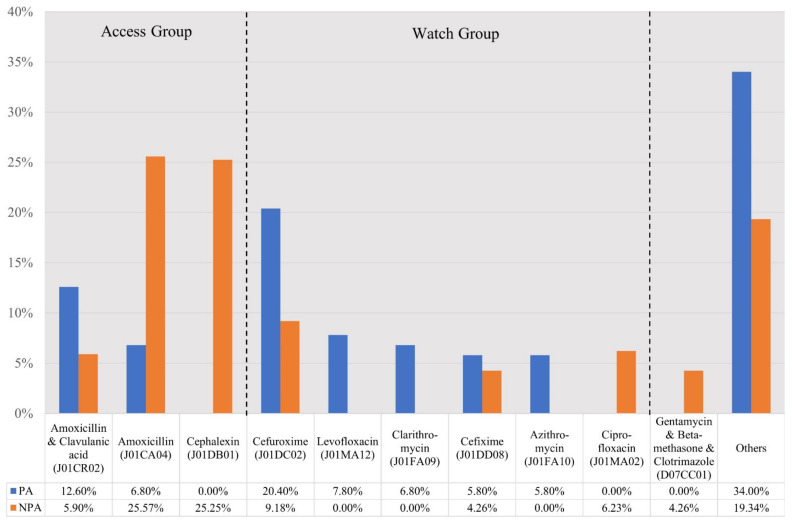
Types of antibiotics purchased with a prescription (PA group) and without a prescription (NPA group). NB. PA: antibiotics purchased with a prescription; NPA: antibiotics purchased without a prescription.

**Table 1 antibiotics-11-01091-t001:** Summary of laws and activities especially among low- and middle-income countries banning the purchasing of antibiotics without a prescription and their impact.

Country	Summary of Initiatives and Their Outcomes
Brazil—private and public pharmacies [33,34,35]	It is generally impossible for public pharmacists in Brazil to sell antibiotics without a prescription—confirmed in the study of Moura et al. (2015) [33]. There was a documented decrease in antibiotic use (1.87 DDD/TID—*p* < 0.001) after restrictions banning the selling of antibiotics without a prescription (2008 to 2012) among private pharmacies in Brazil; this was more prevalent in the more developed regions of Brazil as well as in the state capitals (Moura et al.) [33]. Lopes-Junior et al. (2015) documented decreased sales of amoxycillin (approximately 30%), tetracyclines (30.5% decrease), sulfonamides (28.5% decrease) and macrolides (25% decrease) post legislation despite a general growth in the pharmaceutical market [34]. Mattos et al. (2017) also documented a decrease in the dispensing of antibiotics after legislation was passed, including cephalosporins (−19.4%), quinolones (−12.7%) and aminopenicillins (−11.1%) [35].
Chile [36,37]	Chile was one of the first countries in Latin America to introduce greater enforcement of the law banning the purchasing of antibiotics without a prescription. These activities were enhanced by antibiotics being removed from the list of medicines having sales incentives among pharmacies. Antimicrobial consumption decreased by 31% to 8.5 DID just after the enforcement of the new regulation helped by public information campaigns and enhanced enforcement of the regulations. However, there has been a slow increase in antimicrobial utilization in recent years, suggesting the impact of such laws diminish over time unless pharmacists are continually monitored and additional initiatives introduced when needed.
China [38]	Multiple initiatives have been introduced in Shaanxi Province to reduce the dispensing of antibiotics without a prescription. Measures included stricter regulations for dispensing antibiotics, improving pharmacists’ education with a qualified pharmacist necessarily present to dispense antibiotics, increased frequency of unannounced pharmacy inspections and punishments for misuse. As a result, dispensing of antibiotics without a prescription decreased from 72.3% to 50.2% (*p* < 0·0001) for a 5-year-old child with diarrhoea between 2011 and 2017, and a similar reduction for patients with URTIs —from 95.8% of simulated patients down to 69.5% (*p* < 0·0001).
Colombia [37,39]	The initial enforcement of the law in 2005 had a modest impact on overall sales of antibiotics in the first three years. However, a follow-up study conducted after five years following initial enforcement found a high number of pharmacies (80.3%) were still not complying with the law as a result of lax monitoring. These findings prompted calls for greater enforcement of the law to reduce unnecessary antibiotic consumption.
Mexico [36,40]	The government implemented policies in 2010 to enforce existing laws to reduce the dispensing of antibiotics without a prescription. The new regulations required antibiotic prescriptions to be retained and registered in pharmacies, with fines for non-compliance. As a result, antibiotic utilization decreased by 22.9% between 2007 and 2012, and the trend accelerated after greater enforcement of the legislation. Alongside this, an appreciable seasonal reduction in the consumption of penicillins after greater enforcement of the legislation occurred.
Namibia [41,42]	Pharmacists are aware of current regulations banning the dispensing of antibiotics without a prescription, with their activities regularly monitored. Alongside this, increased education of pharmacists regarding antibiotics and viral infections. A survey among children in households in Namibia with ARIs, including common colds and influenza, found that these children were typically treated with cold/flu medication, decongestants and paracetamol with no dispensing of antibiotics in pharmacies without seeing a physician. A similar situation was seen during the COVID-19 pandemic. Education and monitoring of pharmacies resulted in no change in the antimicrobial utilization patterns during the early stages of the pandemic, assisted by pro-activity among pharmacists, suggesting other potential prevention and management approaches.
Republic of Srpska [43]	Greater enforcement of the regulations banning the self-purchasing of antibiotics in community pharmacies with fines for violations, including EUR 500–1500 for pharmacy directors and EUR 500–750 for pharmacy technicians, along with ongoing activities to try and reduce AMR, which included increased education of pharmacists and the production of guidelines incorporating those for ARIs, decreased the dispensing of antibiotics without a prescription from 58% of requests to 18.5%. In addition, the most common reason for not dispensing an antibiotic for an ARI to a simulated patient after greater enforcement of the regulations was that antibiotics cannot be dispensed without a prescription.
Saudi Arabia [28]	There was greater enforcement of the law concerning the purchasing of antibiotics without a prescription in Saudi Arabia from May 2018 onwards, with fines of up to SAR 100,000 (equivalent to USD 26,666) and cancellations of the license of pharmacists in the case of violations. The purchasing of antibiotics was common before this despite the law (with up to 96.6% of pharmacies dispensing antibiotics to simulated patients with pharyngitis). Following an increase in the penalties, only 12.9% of pharmacists stated that the purchasing of antibiotics without a prescription was still common, with only 12.1% dispensing an antibiotic to simulated patients with pharyngitis and typically only after persistence from patients.
Sri Lanka [29]	Despite legislation banning the purchasing of antibiotics without a prescription, approximately 30% of surveyed pharmacists and pharmacist assistants had supplied antibiotics with a prescription for common infections such as ARIs. However, pharmacists with any form of recognized qualification were less likely to supply antibiotics without a prescription for possible viral infections such as ARIs.
South Africa [44]	Despite legislation, antibiotics for simulated patients with urinary tract infections were dispensed in privately owned pharmacies (80%), although not in corporate (franchised) pharmacies. There was no dispensing of antibiotics for patients with URTIs in any of the pharmacies surveyed. Greater enforcement of the regulations coupled with improved education of pharmacists and their assistants is recommended going forward.
Venezuela [37]	The government implemented policies in an attempt to limit the dispensing of three antibiotic groups without a prescription. However, there were no public awareness campaigns, and the ‘enforcement’ was only via government publications with no follow-up of the regulations among community pharmacists. This resulted in no decrease in antibiotic utilization levels after introduction of the policies. On the contrary, the opposite was observed with an increase in antibiotic utilization.

NB: AMR: antimicrobial resistance; ARIs: acute respiratory infections; URTIs: upper respiratory tract infections.

**Table 2 antibiotics-11-01091-t002:** Demographic characteristics of participants purchasing antibiotics.

Characteristics	N (%)			*p*-Value
	Prescription	Non-Prescription	Total	
**Median age (IQR) (*n* = 470)**	35 (20–78)	35 (18–70)	35 (18–78)	0.511
**Gender (*n* = 480)**				
Male	29 (33.0%)	141 (36.0%)	170 (35.4%)	0.593
Female	59 (67.0%)	251 (64.0%)	310 (64.6%)	
**Levels of education (*n* = 444)**				
High school or lower	44 (55.7%)	254 (69.6%)	298 (67.1%)	0.007
College	12 (15.2%)	58 (15.9%)	70 (15.8%)	
University	23 (29.1%)	53 (14.5%)	76 (17.1%)	
**Occupation (*n* = 465)**				
Freelance work	38 (46.3%)	251 (65.5%)	289 (62.2%)	<0.001
Public sector employee	21 (25.6%)	58 (15.1%)	79 (17.0%)	
Others	23 (28.1%)	74 (19.4%)	97 (20.8%)	

**Table 3 antibiotics-11-01091-t003:** Pathological classification of purchased antibiotics.

Pathological Classification	N (%)		
	Prescription (*n* = 83)	Non-Prescription (*n* = 391)	Total (*n* = 474)
Respiratory	39 (43.8)	260 (65.3)	299 (61.4)
Digestive	12 (13.5)	52 (13.1)	64 (13.1)
Eye	7 (7.9)	22 (5.5)	29 (6.0)
Skin	4 (4.5)	17 (4.3)	21 (4.3)
Genito-urinary	6 (6.7)	8 (2.0)	14 (2.9)
Musculo-skeletal	2 (2.2)	8 (2.0)	10 (2.1)
Ear	2 (2.2)	2 (0.5)	4 (0.8)
Nervous	0 (0)	4 (1.0)	4 (0.8)
Metabolism	2 (2.2)	1 (0.3)	3 (0.6)
Pregnancy	2 (2.2)	1 (0.3)	3 (0.6)
Others	7 (7.8)	16 (4.0)	23 (4.7)

**Table 4 antibiotics-11-01091-t004:** Awareness of customers about antibiotics.

Questions	N (%)			*p-*Value
	Prescription	Non-Prescription	Total	
‘According to you, it is illegal for purchasing of antibiotics without a doctor’s prescription?’	25 (29.8%)	72 (19.2%)	97 (21.1%)	0.032
‘In the future, do you agree to visit the doctor for having a prescription when purchasing antibiotics at the pharmacy/drugstore?’				
*Agree*	56 (63.8%)	160 (44.2%)	216 (48.6%)	0.000
*Partly agree*	17 (20.7%)	127 (35.1%)	144 (32.4%)	
*Disagree/Not sure*	9 (11.0%)	75 (20.7%)	84 (18.9%)	
‘According to you, normally, how long should antibiotics be taken?’	44 (50.6%)	143 (37.6%)	187 (40.0%)	0.026
‘According to you, does not taking antibiotics for long enough lead to antibiotic resistance?’	44 (54.3%)	164 (47.4%)	208 (48.7%)	0.262
‘Have you ever known/heard about “antibiotic resistance”?’	53 (*59.6%*)	200 (*53.2%*)	253 (*54.4%*)	0.279
‘Is antibiotic resistance a serious problem in the community?’				
*Agree*	47 (63.5%)	162 (50.3%)	209 (52.8%)	0.104
*Partly agree*	7 (9.5%)	*33 (10.2%)*	30 (10.1%)	
*Disagree/Not sure*	20 (27.0%)	127 (39.4%)	147 (37.1%)	

**Table 5 antibiotics-11-01091-t005:** Logistic analysis of influential factors for the non-prescribed purchase of antibiotics.

Demographic Characteristic	Univariate Analysis		Multivariate Analysis	
	OR (95% CI)	*p*	OR (95% CI)	*p*
**Gender**				
*Male* *Female*	Ref. 1.143 (0.70–1.87)	0.593		
**Age**				
*<35 years* *≥35 years*	Ref. 0.919 (0.57–1.47)	0.725		
**Occupation**				
*Freelance work* *Others (Non-freelance work)*	Ref. 0.436 (0.274–0.696)	0.000	0.52 (0.83–0.96)	0.017
**Educational level**				
*≤Graduated high school* *College * *University*	Ref. 0.837 (0.416–1.685) 0.399 (0.222–0.716)	0.619 0.002	0.952 (0.51–2.01) 0.494 (0.25–0.96)	0.899 0.039

## Data Availability

The datasets used during the current study are available from the first and corresponding author on reasonable request.

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
