# Peer review of "A National Survey of Dispensing Practice and Customer Knowledge on Antibiotic Use in Vietnam and the Implications"

_antibiotics, 2022, doi:10.3390/antibiotics11081091_

Round 1

Reviewer 1 Report

Abstract:

Line 22; Write AMR in full

Line 23; Write 9 in full.

Is regions the same as provinces? (Compare with line 303).

Line 26; Do not start sentences with numerals.

... who bought ....

About 81.7% ......

Line 30; ... that they ....

Line 32; ... to having a doctor's ....

Line 34; Delete ...highly....

Line 36; ... self-purchasing of antibiotics thus reducing antimicrobial resistance ...

Introduction:

Line 43; ... use thus increasing ....

... is a worldwide ...

Line 44; ... estimated at .....

Line 47; ... costs of up ...

Lines 47-48; Is the GDP referred to here 'global'?

Line 60; How many pharmaceutical regulatory authorities are being referred to?

Normally one regulatory authority operates in a country.

Lines 61-62; ...Vietnam, similar to several other LMICs ....

Line 65; ...in hospital pharmacies after....

Lines 64-72; Rephrase for flow.

Line 73; ... pharmacies should be managed ...

Line 76; ... a middle college diploma ...

Line 89; ... causing concern among ....

Line 90; Delete 'and situations'.

Line 91; .. current status ...

Lines 91-92; This was the rationale ...

Lines 94-97; Rephrase.

Line 101; Delete 'in Vietnam'.

Line 102; ... crucial since they ...

Lines 103-106; Move these statements to the Conclusion section.

Table 1; Table 1 may be omitted. It is more appropriate for a Review paper

not a Research article.

The information in Table 1 can be summarized under the discussion in context

Results:

Line 114; 28.2% is close to quarter (25%) than third.

Line 115; ... antibiotics being dispensed ....

Lines 115-120; Rephrase to remove monotony and flow.

Line 128; Most participants purchasing ....

Line 129; Delete 'most'

Lines 130-131; The group  purchasing .....

Table 2: Levels of education

High school or lower

Middle level college

University

Line 137; Both groups (with or without prescription) ....

Line 140; What is 'over-the-counter antibiotics'?

cough, ...., colds, ....

Line 141; ... throat, ...

Line 143; Antibiotic therapeutic classification of antibiotics studied

Table 2; Column 1 title - ATC

Line 146; ... prescription (29) ...

Line 147; ... prescription (25).

Line 150; ... 29 antibiotics purchased ...

Line 151; .. group and 17 ...

Delete 'and none from the reserve group'

Line 152; ... oral medications.

LIne 154; ... by co-amoxiclav (12.6) ...

Figure 2; Provide clearer chart.

Line 166; ... was illegal in ...

Lines 169-170; ... the prescription group than non-prescription group ...

Insert the percentages appropriately.

Lines 171-175; Reprase for flow.

Table 4; Column 1 - Transcribe the questions verbatim.

Line 178; Participants in the non-prescription group refused ....

Line 180; Delete 'or non-serious'.

Line 181; Delete 'compared with visiting doctors with their conditions'

Line 182; .... their conditions beforehand ...

Lines 185-186; Delete the words in parentheses.

 Line 190; It was noticeable ......

Line 191; ... about AMR.

Lines 193-194; Delete sentence.

Lines 195-196; Abbreviate title.

Line 201; ... lower tendency of .....

Line 203; To the best of our knowledge, this ....

Discussion:

Line 207; ... study provided ....

Line 211; Delete text in parentheses.

Line 213; Delete 'initially'.

Lines 214-232; Rephrase for clarity and flow.

Lines 234-235; Delete 'prior to recent initiatives'.

Line 237; ... these viral infections.

Line 240; ... stipulated fines (... caught).

Lines 240-241; The sentence is incomplete.

Lines 244-247; Rephrase.

Line 252; .... to study findings in ....

Line 255; There is concern regarding 'Watch' ....

Lines 256-257; Delete the words after 'prescrition' in the sentence, but retain the percentage in brackets.

Lines 259-260; Rephrase.

Lines 264-286; Repharse for proper flow. Avoid overuse of 'concern'.

Line 287; Create distinct section for 'Study limitations'.

Line 291; Delete 'we believe'

Lines 294-295; .. the study  instead of directly interviewing them.

Lines 298-299; ... this, the findings provide a robust guidance ...

... Vietnam henceforth.

Materials and methods:

Line 303; Is it provinces or regions? Use consistency terminology.

Line 305; Delete 'in Vietnam'.

Line 312; Delete 'from the current list'.

Line 316; ... drug stores for the study.

Line 318 ... was granted by ...

Line 320; ...., and consent to .....

Line 323: Delete 'carefully'.

Lines 324-325; Delete the words after 'survey' in the sentence.

Lines 327-334; Rephrase for clarity and flow.

Summarize inclusion and exclusion criteria.

Line 338; Customers willing to participate ....

Lines 341-342; What is the meaning of 'outside of the drug retailers'?

Section 4.3; Provide the questionnaire as supplememntary information.

Line 347; .. authors for suitability in the  ....

Line 350; Delete 'slightly'.

Line 352; Delete 'first'.

Lines 356-357; Delete the words after 'customers' in the sentence.

Line 370; Provide authors of the software.

Conclusions:

Lines 387-392; Rephrase for clarity and flow.

Author Response

  1. A) Abstract:

Line 22; Write AMR in full.

Author comments: Now done

Line 23; Write 9 in full.

Author comments: Now done

Is regions the same as provinces? (Compare with line 303).

Author comments: Yes – now changed to provinces throughout to avoid any confusion

Line 26; Do not start sentences with numerals.

Author comments: Now changed

... who bought ....

Author comments: Now changed

About 81.7% ......

Author comments: Now changed

Line 30; ... that they ....

Author comments: Now changed

Line 32; ... to having a doctor's ....

Author comments: Now changed

Line 34; Delete ...highly....

Author comments: Now changed

Line 36; ... self-purchasing of antibiotics thus reducing antimicrobial resistance ...

Author comments: Now changed

B) Introduction:

Line 43; ... use thus increasing ....

Author comments: Now changed

... is a worldwide ...

Author comments: Now changed

Line 44; ... estimated at .....

Author comments: Now changed

Line 47; ... costs of up ...

Author comments: Now changed

Lines 47-48; Is the GDP referred to here 'global'?

Author comments: Now changed

Line 60; How many pharmaceutical regulatory authorities are being referred to? Normally one regulatory authority operates in a country.

Author comments: Now changed

Lines 61-62; ...Vietnam, similar to several other LMICs ....

Author comments: Now changed

Line 65; ...in hospital pharmacies after....

Author comments: Now changed

Lines 64-72; Rephrase for flow.

Author comments: Now amended – hope this is now OK

Line 73; ... pharmacies should be managed ...

Author comments: Now changed

Line 76; ... a middle college diploma ...

Author comments: Now changed

Line 89; ... causing concern among ....

Author comments: Now changed

Line 90; Delete 'and situations'.

Author comments: Now changed

Line 91; .. current status ...

Author comments: Now changed

Lines 91-92; This was the rationale ...

Author comments: Now changed

Lines 94-97; Rephrase.

Author comments: Now updated – hope this is now OK

Line 101; Delete 'in Vietnam'.

Author comments: Now changed

Line 102; ... crucial since they ...

Author comments: Now changed

Lines 103-106; Move these statements to the Conclusion section.

Author comments: Thank you for this. We prefer to also keep these comments here as they lead into the whole rationale for the paper. We hope this is acceptable

Table 1; Table 1 may be omitted. It is more appropriate for a Review paper not a Research article. The information in Table 1 can be summarized under the discussion in context

Author comments: Thank you for this. Other reviewers have commented on why we need this paper. Table 1 really helps to show why we really need this information for the paper, i.e. what is the current situation in Vietnam – especially with a reduction in the possible fines for community pharmacists/ drug stores for breaking the law – and what can be done in the future to address this based on examples primarily in other LMICs. Consequently – Table 1 helps address this (and is why we would like to keep the comments on old lines 103 – 106. We believe Table 1 adds weight to the whole paper – especially as a Table such as this has never been compiled before – building on the review by Jacobs et al (new ref 12). We hope this is now acceptable to you (we firmly believe this paper will be heavily downloaded and cited once published based on the information provided).

C) Results:

Line 114; 28.2% is close to quarter (25%) than third.

Author comments: Now changed

Line 115; ... antibiotics being dispensed ....

Author comments: Now changed

Lines 115-120; Rephrase to remove monotony and flow

Author comments: Now changed

Line 128; Most participants purchasing ....

Author comments: Not sure what is meant here

Line 129; Delete 'most'

Author comments: Changed to the majority

Lines 130-131; The group  purchasing .....

Author comments: Now changed

Table 2: Levels of education - High school or lower; Middle level college; University

Author comments: Now changed

Line 137; Both groups (with or without prescription) ....

Author comments: Thank you for this – but kept as we believe this emphasizes both groups. We hope this is OK.

Line 140; What is 'over-the-counter antibiotics'? cough, ...., colds, ....

Author comments: Now changed

Line 141; ... throat, ...

Author comments: Now reviewed

Line 143; Antibiotic therapeutic classification of antibiotics studied

Author comments: Now changed

Table 2; Column 1 title - ATC

Author comments: Not sure what this means as no ATC grouping here

Line 146; ... prescription (29) ... Line 147; ... prescription (25).

Author comments: Now changed

Line 150; ... 29 antibiotics purchased ...

Author comments: Now changed

Line 151; .. group and 17 ...

Author comments: Not changed as still need to refer to the Reserve Group (see below). Hope this is OK given the need to emphasize this.

Delete 'and none from the reserve group'              

Author comments: Thank you. However – we have chosen to keep this as the Reserve group of antibiotics should only be used in highly selected cases in hospital and we wanted to emphasize this here – hope this is OK.

Line 152; ... oral medications.

Author comments: Now changed

LIne 154; ... by co-amoxiclav (12.6) ...

Author comments: Now changed

Figure 2; Provide clearer chart.

Author comments: Now added in ATC classification as well as made Figure 2 clearer by re-arranging the Figure to concentrate on ‘Access’ antibiotics before moving to ‘Watch’ antibiotics. We hope this is now acceptable.

Line 166; ... was illegal in ...

Author comments: Now changed

Lines 169-170; ... the prescription group than non-prescription group ... Insert the percentages appropriately.

Author comments: Now changed

Lines 171-175; Reprase for flow.

Author comments: Now changed

Table 4; Column 1 - Transcribe the questions verbatim.

Author comments: Now done with the ‘translated’ questions included as supplementary material. Hope this is now acceptable.

Line 178; Participants in the non-prescription group refused ....

Author comments: Now changed

Line 180; Delete 'or non-serious'.

Author comments: Now changed

Line 181; Delete 'compared with visiting doctors with their conditions'

Author comments: We believe we need to keep this as there are co-payments in Vietnam when visiting physicians. In addition, there can be long waits which affects income especially for informal/ casual workers. We hope this is still acceptable. In addition - made comment in the Conclusions about expanding the primary care network of physicians to help out in the future to reduce unnecessary prescribing/ dispensing of antibiotics for essentially viral infections

Line 182; .... their conditions beforehand ...

Author comments: We felt this reflected the comments made – so wish to still keep this if we can.

Lines 185-186; Delete the words in parentheses.

Author comments: Sorry not sure what is meant. We have though slightly changed this to remind authors what we mean. We hope this is OK. We have also deleted the parenthesis from the next paragraph

Line 190; It was noticeable ......

Author comments: Now changed

Line 191; ... about AMR.

Author comments: Now changed

Lines 193-194; Delete sentence.

Author comments: Now changed to make this clearer as there appeared to be a problem.

Lines 195-196; Abbreviate title.

Author comments: Now changed

Line 201; ... lower tendency of .....

Author comments: Now changed

Line 203; To the best of our knowledge, this ....

Author comments: Now changed

D) Discussion:

Line 207; ... study provided ....

Author comments: Now changed

Line 211; Delete text in parentheses.

Author comments: Nothing changed as nothing in parentheses on this line

Line 213; Delete 'initially'.

Author comments: Changed to earlier and expanded as our later study with greatly increased sanctions had an appreciable impact on the purchasing of antibiotics without a prescription (Table 1). We hope this is OK.

Lines 214-232; Rephrase for clarity and flow.

Author comments: Now revised.

Lines 234-235; Delete 'prior to recent initiatives'.

Author comments: Thank you – we would still like to keep this to reflect the earlier and later situations in Saudi Arabia. We hope this is OK.

Line 237; ... these viral infections.

Author comments: Now changed

Line 240; ... stipulated fines (... caught).

Author comments: Thank you – kept the same as we have shown in the Introduction that the fines were reduced in 2013. We hope this is still OK.

Lines 240-241; The sentence is incomplete.

Author comments: Now changed

Lines 244-247; Rephrase.

Author comments: Now changed

Line 252; .... to study findings in ....

Author comments: Now changed

Line 255; There is concern regarding 'Watch' ....

Author comments: Now changed

Lines 256-257; Delete the words after 'prescrition' in the sentence, but retain the percentage in brackets.

Author comments: Now changed to make the sentence clearer

Lines 259-260; Rephrase.

Author comments: Now changed

Lines 264-286; Repharse for proper flow. Avoid overuse of 'concern'.

Author comments: Now changed

Line 287; Create distinct section for 'Study limitations'.

Author comments: Now inserted

Line 291; Delete 'we believe'

Author comments: Now changed

Lines 294-295; .. the study  instead of directly interviewing them.

Author comments: Now changed

Lines 298-299; ... this, the findings provide a robust guidance ...

... Vietnam henceforth.

Author comments: Now changed

E) Materials and methods:

Line 303; Is it provinces or regions? Use consistency terminology.

Author comments: Thank you – used provinces throughout

Line 305; Delete 'in Vietnam'.

Author comments: Now changed

Line 312; Delete 'from the current list'.

Author comments: Now changed

Line 316; ... drug stores for the study.

Author comments: Kept drug retailers as this includes both community pharmacies and drug stores. Hope this is OK.

Line 318 ... was granted by ...

Author comments: Now changed

Line 320; ...., and consent to .....

Author comments: Now changed

Line 323: Delete 'carefully'.

Author comments: Now changed

Lines 324-325; Delete the words after 'survey' in the sentence.

Author comments: Now changed

Lines 327-334; Rephrase for clarity and flow.

Author comments: Now updated

Summarize inclusion and exclusion criteria.

Author comments: Now changed

Line 338; Customers willing to participate ....

Author comments: Now changed

Lines 341-342; What is the meaning of 'outside of the drug retailers'?

Author comments: Now changed

Section 4.3; Provide the questionnaire as supplememntary information.

Author comments: Now done – we will ask the Journal for a URL link to the translated questionnaire when and if the paper is published.

Line 347; .. authors for suitability in the  ....

Author comments: Now changed

Line 350; Delete 'slightly'.

Author comments: Now changed

Line 352; Delete 'first'.

Author comments: Now changed

Lines 356-357; Delete the words after 'customers' in the sentence.

Author comments: Now changed

Line 370; Provide authors of the software.

Author comments: Now inserted

F) Conclusions:

Lines 387-392; Rephrase for clarity and flow.

Author comments: Now changed to provide additional guidance building on the examples and their impact in Table 1. This also includes a greater role for family physicians with the potential for remote consultations and electronic prescribing. We hope this is acceptable.

Author Response

The purported purpose of this mixed methods research was to assess non-prescription antibiotic sales in Vietnam by “comprehensively investigating antibiotic dispensing practices on a national scale in Vietnam as well as evaluating customer knowledge on antibiotic use. Customers’ knowledge is seen as crucial as they are a key driver of antibiotic prescribing and dispensing in ambulatory care, especially for self-limiting conditions such as URTIs. The findings can be used to direct the authorities in Vietnam regarding potential appropriate and effective interventions to help reduce the extent of inappropriate dispensing of antibiotics in the community. This approach builds on the experiences among other countries, especially LMICs, with their laws banning the purchasing of antibiotics without a prescription.” Much of the data is in line with earlier findings and hence the novelty value of the research findings in this work is low [1-6].

Author comments: Thank you for this. Whilst this study builds on others, as mentioned in the Introduction, we do mention previous studies in Vietnam (building on provided refs 1,2,4 and 6 – some of which are very old). In addition, building on Ref 3 which is now very old with Auta et al and others. We are also unaware of other studies that have brought together a wide review of potential interventions to reduce inappropriate self-purchasing of antibiotics and their impact to provide guidance to a country that has made this practice illegal with fines – but still appreciably happens. For instance Jacob et al (which we mention) discusses potentially banning measures and their impact – but did not look at e.g. the impact of increased education of pharmacists (as seen in Kenya) or a combination of greater restrictions and education of pharmacists as seen in the Republic of Srpska. In addition, the extent of fines and their impact as seen in e.g. Saudi Arabia – which would though be difficult in Vietnam. The study that mentions Sri Lanka (which is now included) talks about increased education for personnel in community pharmacies but not their impact (this study though adds weight to the findings in Vietnam). However, we have included the findings in Table 1. This is why we believe this study is novel providing direction to the authorities in Vietnam – especially with decreasing fines in recent years for illegal selling of antibiotics. These issues are discussed further in the revised conclusion and recommendations going forward. As a result, we believe this study is novel providing future direction not only to the authorities in Vietnam but other LMICs wrestling with similar issues. Consequently, we believe it will be extensively downloaded and cited.

Major comments

There are some major issues which need to be addressed by the authors:

Comment #1: Methods

4.2. Ethical considerations, data collection and sample population and data collection: Who were the data collectors in this study?

Author comments: This is included in Methodology, i.e. fully trained fifth-year pharmacy students and a number of the co-authors

Who all contributed to data analysis & writing the research manuscript? Clearly state these.

Author comments: Thank you – this is clearly stated in Author contribution section at the end of the paper. We hope this is OK.

This study says…“Data were collected over a period of two years by fifth-year pharmacy students and a number of the co-authors from the Hanoi University of Pharmacy, who were carefully trained before the onset of the survey to enhance the output as well as make sure the data collectors were mindful of the sensitivities of the situation.” Would’nt this lead to data collection and reporting biases ?

Author comments: Thank you so much for your comments. We beg to differ though as we believe data collection and reporting were not biased because the data collection process was supervised and managed daily and directly by researchers, who were also the authors of this study and had research experience of at least ten years in community studies. Besides, data collectors were trained and tested carefully by these researchers beforehand. Moreover, the data were collected via the structured questionnaire, and there were no changes in pharmacy policy that affected the collected data. We hope this is acceptable.

Comment #2: Methods

4.2. Ethical considerations, data collection and sample population and data collection:

Author comments: Thank you so much for this. We would like to response each question one by one as follows:

How were rural and urban centres selected in this study for adequate representation?

Author comments: As seen in the methodology, the sample selection was described carefully in the study setting. We hope this is now acceptable

What were the average number of clients/pharmacy/day ?

Author comments: Thank you – now added in.

Did this vary in rural and urban settings ?

Author comments: The study design and the questionnaire did not vary by setting. However – as discussed in the Introduction community pharmacies are more common in urban setting and drug stores in rural settings. We hope this is now clear.

What specific type of interviews were conducted ? What were the themes of these interviews to identify important causes of inappropriate antibiotic dispensing ? Were all contents of the conversations recorded and were transcripts made and translated into English ? Were data from transcripts analyzed using qualitative content analysis by listening to the tapes and reading and re-reading the transcripts to become familiar with the data and to categorize information ? How were the common themes identified ? How were the connections within and between themes identified ? Could the researchers include some narrative aspects from the recorded transcripts to make this study more valuable and impactful since this appears to be a mixed-methods study ? This current version of the research manuscript has failed to document these issues.

Author comment: Thank you for this comment. We have now added information about the questionnaire to make the method clearer. As seen, we used a structured questionnaire for the study conducted by face-to-face interviews. In addition, our study was a quantitative study so comments relating to data transcription, themes, qualitative content analysis, are narrative aspects from the recorded transcripts are just not pertinent. We hope this is OK. A translated version of questionnaire is available as Supplementary material.

Comment #3: Methods

4.3. Study questionnaire:

Author comments: Thank you so much for your comments. We would like to response each question one by one as follows:

How did the study researchers assess the reliability of survey responses ?

Author comments: Thank you for this. As mentioned, the survey was quantitative in nature limiting the opportunities for any misunderstandings. Otherwise – this is an issue with any questionnaire. We hope this is now OK.

How were answers to the questions reported ? Likert scale or otherwise ? Was Cronbach’s alpha analyzed with respondents’ scores for all questionnaire items ?  What cut-off value was accepted for consistency ? Were all forms anonymized to encourage interviewees to frankly share information?  Clearly state all these issues raised.

Author comment: Thank you for this comment. As mentioned, a face-to-face interview structured questionnaire was used in the survey. The specific contents of the questionnaire were also shown in Table 4. The first four questions were answered yes/no/unknown and the following two questions were answered on a Likert 3-point scale and unknown. That’s why the issues relating to Cronbach’s alpha, cut-off value, and forms were not applied in our study. We hope this is OK. We have also commented on the anonymised forms.

Comment #4: Results: Figure 2: Since, antibiotics were classified by the Anatomical Therapeutic Chemical (ATC) Classification System, and subsequently by the WHO AWaRe antibiotic classification, such data need to be included in this Figure.

Author comments: Thank you – now updated in the revised manuscript

Comment #5: Discussion

Could the causes for irrational/illegitimate antibiotic purchases be included in a tabular format with frequencies ? The authors must also clearly highlight what new and novel findings they wish to communicate in this research paper and how it is different from factors reported elsewhere and in other similar studies. Please expand the section on proposed solutions.

Author comments: Thank you for this comment. We have highlighted in the Introduction the multiple causes for irrational/ illegitimate antibiotic purchases. We have also included a novel Table (Table 1) of different methods used across countries to try and reduce the levels of self-purchasing ranging from education to more stringent measures – in addition highlighted with e.g. examples in South America that constant attention is needed (novel to combine a number of these building on e.g. Jacobs et al – mentioned in the references) – with a greater focus on LMICs. We have used this knowledge to suggest future activities for the MoH in Vietnam compared to general comments on increasing knowledge levels of pharmacy staff in the paper by Zahwir et al (Sri Lanka). We hope this is acceptable.

Comment #6: English language needs touching up in a major way. The article needs to be rewritten in readable English. Many sentences are confusing, do not lead to scientific meaning.

Author comments. Thank you for this. The manuscript has now been updated by one of the co-authors who is a native English speaker with over 450 publications in peer-reviewed Journals. We hope this is now acceptable.

References:

  1. Nga DT, Chuc NT, Hoa NP, Hoa NQ, Nguyen NT, Loan HT, Toan TK, Phuc HD, Horby P, Van Yen N, Van Kinh N. Antibiotic sales in rural and urban pharmacies in northern Vietnam: an observational study. BMC Pharmacology and Toxicology. 2014 Dec; 15(1): 1-10.
  2. Chuc NT, Tomson G. “Doi moi” and private pharmacies: a case study on dispensing and financial issues in Hanoi, Vietnam. European Journal of Clinical Pharmacology. 1999 Jun; 55(4): 325-32.
  3. Morgan DJ, Okeke IN, Laxminarayan R, Perencevich EN, Weisenberg S. Non-prescription antimicrobial use worldwide: a systematic review. The Lancet Infectious Diseases. 2011 Sep 1; 11(9): 692-701.
  4. Nguyen HH, Ho DP, Vu TL, Tran KT, Tran TD, Nguyen TK, van Doorn HR, Nadjm B, Kinsman J, Wertheim H. “I can make more from selling medicine when breaking the rules”–understanding the antibiotic supply network in a rural community in Viet Nam. BMC Public Health. 2019 Dec; 19(1): 1-11.
  5. Zawahir S, Lekamwasam S, Aslani P. Factors related to antibiotic supply without a prescription for common infections: a cross-sectional national survey in Sri Lanka. Antibiotics. 2021 May 28; 10(6): 647.
  6. Larsson M, Kronvall G, Chuc NT, Karlsson I, Lager F, Hanh HD, Tomson G, Falkenberg T: Antibiotic medication and bacterial resistance to antibiotics: a survey of children in a Vietnamese community. Trop Med Int Health 2000, 5(10): 711–721.

Reviewer 3 Report

Dear Authors,

The manuscript entitled “A National Survey of Dispensing Practice and Customer Knowledge on Antibiotic Use in Vietnam and the Implicationsis analyses of frequency of antibiotic use, their source, knowledge of the customers about drugs and why they buy them. This a very current and original article which is touching important issue of antibiotic abuse in Vietnam. In era of multi drug resistance it essential to carry about antibiotic use and provide full control.

The paper require some corrections:

    1. The whole manuscript is good written but organized in the wrong order. It should be as always - Introduction, Materials and Methods, Results, Discussion, Conclusions and References. Introduction is very long and contains general data but the Authors wanted to gave proper background as I assume.

    2. The aim of the work should be clearly said in the paper at the end of Introduction.

    3. Line 141 – would be better to use term flu-like symptoms

    4. Line 236 – one again suggestion about symptoms – better to use <when it was observed flu like symptoms, and symptoms of cold.

      Cough is unnecessary because it is the symptom of cold.

Based on the results, an interesting discussion was prepared and adequate conclusions were drawn. The Authors wrote about limitations of their work which is very important.

I think, it is very interesting and very valuable article, describing problem which somehow is hidden in many countries.

Author Response

The manuscript entitled “A National Survey of Dispensing Practice and Customer Knowledge on Antibiotic Use in Vietnam and the Implications” is analyses of frequency of antibiotic use, their source, knowledge of the customers about drugs and why they buy them. This a very current and original article which is touching important issue of antibiotic abuse in Vietnam. In era of multi drug resistance it essential to carry about antibiotic use and provide full control.

The paper require some corrections:

  1. The whole manuscript is good written but organized in the wrong order. It should be as always - Introduction, Materials and Methods, Results, Discussion, Conclusions and References. Introduction is very long and contains general data but the Authors wanted to gave proper background as I assume

Author comments: Thank you – the layout of the chapters is dictated by the Journal – hopefully acceptable! We agree that the Introduction is long. However, we not only wanted to set the scene in Vietnam – which is different to a number of other LMICs – but also provide a comprehensive overview of activities undertaken by governments to reduce the self-purchasing of antibiotics without a prescription. We felt that this was very important for this paper as the authorities in Vietnam actually reduced the level of fines, and have not updated these since 2013 in line with inflation. Consequently – perhaps not surprising – still see high levels of self-purchasing of antibiotics in Vietnam. The findings in Table 1 also played a key role in our suggestions for the future. Consequently – we feel that its length is fine. We hope you agree.

  1. The aim of the work should be clearly said in the paper at the end of Introduction.

Author comments: Thank you – now added in a section on study objectives at the end of the Introduction. We hope this is acceptable.

  1. Line 141 – would be better to use term flu-like symptoms.

Author comments: Thank you – now done.

  1. Line 236 – one again suggestion about symptoms – better to use <when it was observed flu like symptoms, and symptoms of cold. Cough is unnecessary because it is the symptom of cold.

Author comments: Thank you – now amended where we can. We understand about a cough – but this paper is also targeted at a more general audience. We hope this is acceptable.

  1. Based on the results, an interesting discussion was prepared and adequate conclusions were drawn.The Authors wrote about limitations of their work which is very important.

I think, it is very interesting and very valuable article, describing problem which somehow is hidden in many countries.

Author comments: Thank you for these kind comments - appreciated. We have also expanded the Conclusion and Recommendations to add further value to the paper. We hope this is OK!